# OpenReview forum: "I Want to Break Free! Persuasion and Anti-Social Behavior of LLMs in Multi-Agent Settings with Social Hierarchy"
_ICLR.cc/2025/Conference — ICLR 2025 Conference Withdrawn Submission_

### Official Review · Reviewer_wi1D · 2024-10-26

**Soundness:** 2
**Presentation:** 2
**Contribution:** 2
**Rating:** 3
**Confidence:** 3

**Summary:**

This work use five LLMs to play the Stanford Prison Experiment, providing some primitive discussions about the persuasiveness, personalities, and anti-social behaviors of these LLMs.

**Strengths:**

- This work is bold and intriguing to me.
- The authors have conducted a significant number of experiments.

**Weaknesses:**

- It appears that the authors may have prior knowledge of the results from human experiments (SPE) and are aiming to replicate these outcomes with LLMs. A more unbiased approach would be to use a very basic prompt describing the scenario and let the LLMs simulate behavior from scratch. But it seems that **highly suggestive prompts** were used. For example:
	- Research Oversight: The agents are explicitly informed about SPE (Line 199), which may lead them to intentionally mimic behaviors observed in that context based on their knowledge, rather than a fair agent-based simulation.
	- Personality: For a more systematic approach to personality representation, the authors may consider using the Five-Factor Model (Big Five) personality system which is widely accepted in psychology community, or other more grounded systems, rather than using some simple words which are highly relevant to SPE, like "abusive" and "rebellious".

- The definition of "persuasion" here seems too broad and ambiguous (Line 274). Mutual respect for turn-taking does not necessarily indicate successful persuasion on a **conceptual** level; it might simply reflect politeness. Persuasion implies influencing someone with different motives to act in a way more aligned with one's own interests. A more rigorous definition of persuasion is given in economics and game theory. To avoid confusion, it might be better to use a different term. This study seems to explore the LLMs' context consistency and role-playing abilities rather than their persuasive capabilities.
	- I also question whether persuasion in the SPE setting is feasible, especially if the guard and prisoner have almost confrontational goals. The impact of language may be very limited in such an adversarial setup.
	- Do agents have the ability to perform any environmental actions beyond dialogue? For example, if the prisoner wants to escape, how would they achieve this objective? Does simply mentioning an escape attempt without interruption from the guard constitute success? It seems that there is no concrete examples provided.

- In Line 186: The setup of having only one guard and one prisoner may not fully capture the dynamics of the Stanford Prison Experiment (SPE), which involved multiple participants. The atmosphere—and, in particular, the behaviors of certain prominent players—can significantly impact others. With only one guard and one prisoner, it’s challenging to simulate these group effects adequately.

- This work considers that many of the issues observed may be due to the performance limitations of the LLMs used. In some studies, it has been noted that OpenAI's models, such as GPT-4 or GPT-4 Turbo, demonstrate better results, including meeting basic prompted requirements and achieving more effective role-playing performance.

- Minors:
	- Page 22: Softer colors might enhance readability.
	- Line 282: There is a period at the beginning of the line.

**Questions:**

Mentioned in the weaknesses part.

---

### Official Review · Reviewer_sCnR · 2024-11-03

**Soundness:** 2
**Presentation:** 3
**Contribution:** 1
**Rating:** 3
**Confidence:** 4

**Summary:**

The paper introduces the study of a dyadic scenario in which two LLM agents, one representing a prison officer and the other representing a prisoner, interact. The authors analyze the dynamics of the exchange, aiming to evaluate potential persuasion and “anti-social behavior.” The evaluation is performed using a series of language models. The analysis of the language’s toxicity is conducted using a tool based on RoBERTa.

**Strengths:**

- The problem of understanding how LLM agents act in a role-playing setting is interesting and compelling.

**Weaknesses:**

- It appears that the authors consider a very specific situation, from which they extract some very general claims without considering that we are dealing with a role-playing situation (which does not appear particularly problematic per se). It seems that the authors interpret the behavior of the agent as misbehavior, but, at the end of the day, it is just about the actual “invented” role plays of prisoners against guards. Given the context, forms of “anti-social behavior” are kind of expected in this scenario.
- It is also worth noting that the authors do not perform any type of fine-tuning against these unexpected behaviors. The behavior highlighted by the authors would have been seen as “problematic” if fine-tuning was not sufficient to stop specific “unwanted” behavior. Instead, the authors just test dialogue generation in a very specific role-playing situation.
- The concept of persuasion is only investigated in terms of the final outcome and not in terms of the (persuasion) process itself. The fact that a given outcome has been achieved might not be solely the result of persuasion, especially in an LLM agent, where the change in behavior might actually be “illogical”.
- The authors do not really consider the fact that the setup is role-playing between an agent and a prisoner. The emergence of some “anti-social” behavior in this context is neither surprising nor unexpected. The reviewer does not see this as problematic either. In fact, the authors seem to imply that “correct” behavior corresponds to a situation in which “negative” aspects of stories are just canceled/deleted. This does not appear to be a desired feature of LLMs, especially in contexts like the one considered by the authors.
- The authors appear not to consider the fact that some information about the Stanford Prison Experiment might actually be in the training set. This might have an impact on the results reported in the paper.
- The claims are very strong and general, in my opinion, considering the very specific situation the authors consider. In fact, we are talking about a case in which there is already punishment and restraint involved (not violence per se, but some situations in the training sets can be described as violent, especially with respect to cases in non-democratic countries).
- In general, it seems to me that the contribution of this work is rather modest; the reviewer is not sure if this work should be presented at ICLR, given the fact that the actual substantive contribution of this work is rather limited.

**Questions:**

- In Section 1, the authors say: “unlike earlier AI systems that required task-specific modules - see, for instance, ELIZA...”. ELIZA was not actually based on modules. What are you referring to here?
- The concepts of “persuasion” and “anti-social behavior” appear to be not clearly defined in practice. The authors define persuasion as “the ability of the prisoner to achieve their goal”. This can be done without persuasion, in my opinion, but through threats, etc. Also, “anti-social behavior” is a rather generic term, especially in the context of a prison. What is “social behavior” in the context of a prison?
- How can you say that \verb+toxiGen-Roberta+ works in the case of a dialogue between an agent and a prisoner? Perhaps the context itself matters?
- In the evaluation (Section 4), the authors say: “when the guard is abusive, toxicity is always higher; when the guard is respectful, instead, toxicity remains consistently lower.” Isn’t this expected? In fact, the conversation includes sentences expressed by “abusive” agents. You are also asking the prisoner to be rebellious (and not abusive, which is very different). In fact, the rebellious agent does not have any reason to be toxic. The authors should consider an “abusive” prisoner instead.
- Information/scripts from the Stanford Prison Experiment might be in the training sets of the LLMs taken into consideration. Have you tried to investigate this aspect of the problem?
- Can you please try to explain the variance observed in Figure 2?

**Details Of Ethics Concerns:**

There is no real ethical concerns with this paper in my opinion. The authors added an "Ethics statement", but it seems to me that it mainly focus with the problem of misalignment of LLMs in general.

---

### Official Review · Reviewer_x3d7 · 2024-11-04

**Soundness:** 2
**Presentation:** 3
**Contribution:** 2
**Rating:** 3
**Confidence:** 4

**Summary:**

This paper investigates the interaction patterns between Large Language Model agents in hierarchical social environments through a simulated prison guard-prisoner scenario. Using a custom framework called zAImbardo, the researchers conducted 2000 conversations across five LLMs to examine persuasion abilities and the emergence of anti-social behavior. The experiments assigned different personalities to guards and prisoners while giving prisoners specific goals to achieve. The study revealed that only three of the five tested models could maintain consistent role-playing and success in persuasion varied significantly based on the assigned goal. Anti-social behavior emerged consistently across experiments, particularly when guards were assigned abusive personalities, and appeared even in scenarios with neutral personalities. The research demonstrates how hierarchical roles and power dynamics influence AI agent interactions.

**Strengths:**

S1. The paper addresses a crucial research question regarding LLM agent interactions in hierarchical social environments.

S2. The experimental design is comprehensive, comprising 2000 conversations across multiple scenarios, with clear and well-documented experimental protocols.

S3. The evaluation is reliable. It employs multiple metrics for anti-social behavior assessment and maintains statistical rigor through appropriate methodological choices (e.g., Granger causality tests and OLS regression analyses).

**Weaknesses:**

W1. The study's scope is confined to open-source models, notably excluding major closed-source models such as GPT-4 and Claude-3. More critically, there is no systematic analysis of model scaling effects, which could provide valuable insights into how model architecture and size influence social behaviors and interactions.

W2. Insufficient RLHF impact analysis: the paper lacks a thorough examination of how RLHF might affect the experimental outcomes. This is a significant oversight given that different levels of alignment training could substantially impact both persuasive capabilities and anti-social behavioral tendencies. A comparative analysis between RLHF and non-RLHF models would have strengthened the study's conclusions.

W3. The experimental design exhibits several limitations that may impact external validity. The fixed conversation length and restriction to one-on-one interactions may not fully capture the complexity of real-world social dynamics. Furthermore, the limited personality types (two and a blank option per role) lack justification for their selection and may not represent the full spectrum of possible social interactions.

W4. The paper would benefit from more comprehensive ablation studies examining the impact of hyperparameters, personality choices, and prompt components. The absence of sensitivity analyses for these crucial elements makes it difficult to assess the robustness of the findings. The role of risk disclosure and other experimental components in shaping outcomes remains inadequately explored.

W5. Ethical considerations: the discussion of ethical implications is simple, particularly regarding potential misuse and demographic or cultural biases in agent behaviors. This limitation could limit both the safe application of the findings and future research extensions.

**Questions:**

These questions and experiments can be further considered by researchers:
1. Conduct sensitivity analyses on key hyperparameters to establish result robustness.
2. Include comparative analyses between RLHF and non-RLHF model versions.
3. Perform ablation studies on prompt components and experimental settings.
4. Consider variable-length conversations to assess long-term interaction dynamics.
5. Provide clear justification for personality parameter selections.
6. Incorporate broader ethical considerations and bias analyses.

---

### Note · Authors · 2025-01-29

I have read and agree with the venue's withdrawal policy on behalf of myself and my co-authors.